# Baselining Urban Ecosystems from Sentinel Species: Fitness, Flows, and Sinks

**DOI:** 10.3390/e27050486

**Published:** 2025-04-30

**Authors:** Matteo Convertino, Yuhan Wu, Hui Dong

**Affiliations:** 1Ecosystem Intelligence & Design Center (TREES), and Nature-Positive Design Hub (N+D), Institute of Environment and Ecology, Tsinghua Shenzhen International Graduate School (SIGS), Tsinghua University, Shenzhen 518055, China; wu-yh22@mails.tsinghua.edu.cn; 2Shenzhen Key Laboratory of Ecological Remediation and Carbon Sequestration, Tsinghua Shenzhen International Graduate School (SIGS), Shenzhen 518055, China; 3Fairy Lake Botanical Garden, Chinese Academy of Sciences, Shenzhen 518055, China; faye.huidong@gmail.com

**Keywords:** butterflies, ecosystem fitness, habitat suitability, ecological flows, attraction basins, parks

## Abstract

How can the shape of biodiversity inform us about cities’ ecoclimatic fitness and guide their development? Can we use species as the harbingers of climatic extremes? Eco-climatically sensitive species carry information about hydroclimatic change in their distribution, fitness, and preferential gradients of habitat suitability. Conversely, environmental features outside of the species’ fitness convey information on potential ecological anomalies in response to extremes to adapt or mitigate, such as through urban parks. Here, to quantify ecosystems’ fitness, we propose a novel computational model to extract multivariate functional ecological networks and their basins, which carry the distributed signature of the compounding hydroclimatic pressures on sentinel species. Specifically, we consider butterflies and their habitat suitability (HS) to infer maximum suitability gradients that are meaningful of potential species networks and flows, with the smallest hydroclimatic resistance across urban landscapes. These flows are compared to the distribution of urban parks to identify parks’ ecological attractiveness, actual and potential connectivity, and park potential to reduce hydroclimatic impacts. The ecosystem fitness index (EFI) is novelly introduced by combining HS and the divergence of the relative species abundance (RSA) from the optimal log-normal Preston plot. In Shenzhen, as a case study, eco-flow networks are found to be spatially very extended, scale-free, and clustering for low HS gradient and EFI areas, where large water bodies act as sources of ecological corridors draining into urban parks. Conversely, parks with higher HS, HS gradients, and EFIs have small-world connectivity non-overlapping with hydrological networks. Diverging patterns of abundance and richness are inferred as increasing and decreasing with HS. HS is largely determined by temperature and precipitation of the coldest quarter and seasonality, which are critical hydrologic variables. Interestingly, a U-shape pattern is found between abundance and diversity, similar to the one in natural ecosystems. Additionally, both abundance and richness are mildly associated with park area according to a power function, unrelated to longitude but linked to the degree of urbanization or park centrality, counterintuitively. The Preston plot’s richness–abundance and abundance-rank patterns were verified to reflect the stationarity or ecological meta-equilibrium with the environment, where both are a reflection of community connectivity. Ecological fitness is grounded on the ecohydrological structure and flows where maximum HS gradients are indicative of the largest eco-changes like climate-driven species flows. These flows, as distributed stress-response functions, inform about the collective eco-fitness of communities, like parks in cities. Flow-based networks can serve as blueprints for designing ecotones that regulate key ecosystem functions, such as temperature and evapotranspiration, while generating cascading ecological benefits across scales. The proposed model, novelly infers HS eco-networks and calculates the EFI, is adaptable to diverse sensitive species and environmental layers, offering a robust tool for precise ecosystem assessment and design.

## 1. Introduction

### 1.1. Computational Eco-Complexity Approach to Ecosystem Fitness

As climate variability increases and global temperatures rise, communities face more frequent and severe events, such as hurricanes, heat and cold waves, floods, and droughts, which are compounding risks linked to the same eco-hydrological dysbiosis [1]. The need for ecosystem predictions, anchored into synthesized species and environmental information, allows stakeholders to support ecosystem planning, design, and engineering critical for communities, ecosystems (natural and built environments together), and economies.

Biocomplexity or ecological complexity [2] responds to these needs with a consequential engineering focus aiming to extract the salient data to construct models and/or information with strong predictability for patterns of interest. Although models and/or information are supportive of decision-making (see Li and Convertino [3] and Riva et al. [2]) and directly useful for eco-hydrology-based ecosystem sensing, adaptation, climate change mitigation, and creation of new ecosystems, all activities are synthesizable as “eco-enhancement” [1,4,5]. In this perspective, ecosystem health (or fitness measured by the habitat suitability) should not be interpreted only as a function related to the ecological dysbiosis of species and/or communities (e.g., of symptoms) but the ecological function of ecosystems manifested by habitat response to hydroclimatic pressure [1,6] (or vice versa concerning the water-carrying capacity in a hydrological focus [7]). The latter is associated with the foundational eco-hydrological architecture of ecosystemic function and its equilibrium with the environmental forcing (i.e., the envirome, see Li and Convertino [8]), leading to optimal ecological patterns that likely support desired ecosystem services (see Convertino and Valverde Jr [1]). This is, for instance, the case of species speciation and dispersal, which lead to the formation of biodiversity patterns and cascading ecosystem services. The core eco-hydrological structure of ecosystems, as coordinated corridors and flows, is the basis for the healthy/functional feedback of all species, communities, and climates (see Wang and Convertino [9] and Zhang and Convertino [10] for coastal ecosystems). Departure from the eco-environmental equilibrium leads to ecosystemic stress [8]. Ecological patterns self-emerge from the organization of eco-hydrological networks (topologies) [3], shaping punctuated evolution toward optimal patterns such as the log-normal Preston’s plot [11]; the divergence of patterns informs us about the shift in ecological fitness from optimality (associated with organized networks such as scale-free and small-world networks [8,12]), including the potential systemic dysbiosis [1]. Ecosystem health not only predicts diseases in populations and communities but it also maps and engineers the baselining eco-environmental foundations of all species and habitat functions. Therefore, this study deliberately aims to increase the eco-hydrological fitness of ecosystems in a way that positive species–habitat–climate feedback is reinforced, and negative cascading outcomes such as biodiversity loss and hydrological extremes are minimized [1,4,5]. To support this effort, the synthesis of ecological information through information- and network-theoretic models (with no assumptions on mechanisms [3]) is necessary for collective decision-making, i.e., *ecological intelligence* from analysis to information actuation. This is aligned with the concept of essential biodiversity variables (EBVs), which was introduced to advance the collection, sharing, and use of biodiversity information [13,14,15] coupled with Earth observations, providing a way to aggregate the many biodiversity observations collected through different methods such as in situ monitoring or remote sensing, to support decisions.

In this study, specifically, we aim to reconstruct the spatial patterns of butterfly habitat suitability where suitability can be intended as the ecological fitness of the community/habitats where butterflies live. This is based on previous studies that investigated butterflies as sentinels for ecosystems [16,17,18,19]. The model that is the focus of this proof-of-concept study is neither an ecological investigation into butterflies nor something applicable only to one species: the paper aims to provide a model for assessing multicriteria fitness, ecological flows, and sinks based on previously developed models. These models are vastly grounded on information-/network-theory and statistical physics (see Li and Convertino [3], for instance). Additionally, the study has a keen interest in the eco-hydroclimatological underpinnings of habitat suitability; as a result, all data are gathered concerning information regarding the reconstruction of eco-hydrological networks, maximizing the predictability of habitat suitability patterns. The ultimate goal is to pave the way for new models supporting urban ecosystem assessment and design (“Digital Ecosystem Models”, DEM, whose inputs and outputs can be improved with further information, including other species, to be tested against the constrained predictability), and not to provide models reproducing species or community dynamics mechanistically.

### 1.2. Species as Indicators of Ecosystem Fitness Stressed by the Climate: Inferring Magnitude and Pathways of Change

Can we use ecological patterns to assess the ecosystemic fitness of a city? Can we use sensitive species to human-driven eco-hydrological change to evaluate the performance of the natural and built environment, such as urban parks? Can we infer these species’ eco-flows and their convergence with eco-hydrological corridors to optimize ecosystem design? Are these flows early sentinels of climate stress? Shapes of ecological patterns are a reflection of climate stressors. In this perspective, we aim to propose a conceptual model to untangle the key network features of ecological patterns (eco-flow topology and motifs) and attribute their variability to the underlying environmental determinants (climatic) constrained to eco-hydrological corridors in cities.

From outputs (e.g., eco-patterns and their change, i.e., systemic stress), the aim of causal inference is to reconstruct input determinants probabilistically: our work is focused on eco-networks and flows (that is, considering how the inputs are structured), and their change from optimal, random, or alternative known networks. Inverse modeling helps us understand the causal function between inputs and outputs, and that function is used in inversion to calculate the unknown parameters linking input–output changes. Inverse modeling is often used when the forward model (from cause to effect) is known, and we want to reverse it, such as through Monte Carlo filtering in global sensitivity and uncertainty analyses. Inference, instead, is a data-based reconstruction of structure–function causal relationships (of causes and effects, such as climate and species [9,20,21,22]) that cannot be known a priori or modeled by mechanistic equations; therefore, the inference is useful to derive the analytics of critical and predictable dynamics, indicators, and system response intensity. Thus, eco-environmental computing (e.g., via MaxEnt, which is a low-level machine-learning model), in a broader information flow perspective [23,24] (where the inferred flow has a physical and computing meaning), is not just about learning the parameters but learning all functional forms of potential species information processing [3]. Learning the precise physics is particularly daunting and maybe utopian for highly complex systems with many components/variables interacting non-linearly (due to space–time delays and the stress–strain non-linear response). Inferred networks, because of the inference construction maximizing traceability, are the most predictive for observed ecological patterns [3]. However, we also aim to provide *causal physicality* to inferred networks: networks encode eco-hydrological networks such as species dispersal networks and hydrological flow networks (in natural landscapes and green corridors) that are dependent on each other and impact climate. The inferred networks also contribute to improving theories of meta ecosystems for predicting systemic function and risks in relation to essential drivers [25,26].

In this modeling framework, minimum resistance networks (or maximum information flow/maximum prediction as in the optimal information network (OIN) model of Servadio and Convertino [27] and Li and Convertino [3]) along the maximum gradient of a multicriteria function (defining the Habitat Suitability, HS) allow us to identify paths where species can spread (as “waves”) in a suitable climate range. These networks, which are typically missing from current HS models [28,29], have the highest predictability of ecological patterns, such as HS. The flow on these networks is related to the accumulation of the minimum resistance and can inform us about an area’s attractiveness (cumulated flow) to species and uncertainty reduction. We pay particular attention to a temperature range that is very similar to the one for humans [30], and the question is whether species-related multicriteria climate waves (or any other climate-sensitive species) coincide with eco-hydrological corridors. Based on the questions and the idea of using species as eco-environmental indicators of suitable habitats for humans, we analyze the situation in Shenzhen, which is one of the largest and rapidly changing megalopolis worldwide. Specifically, we consider butterflies as optimal eco-indicators of ecological community fitness and optimal temperature range. In addition to ecological flows, we analyze how macroecological features, such as abundance and local richness, arise from the distribution of ecological flows and corridors (including longitudinal gradients reflecting urbanization trends and environmental protection) and how this information can be used for the assessment and design of urban parks or similar “green/nature-based solutions” (e.g., green buildings, constructed wetlands, etc.). The assessment of urban parks is carried out by predicting the potential robust association between butterflies and climate features and analyzing that association conditional to urban parks.

Butterflies are potentially good eco-indicators for eco-hydrological change in ecosystems, although they are largely under-reported in terms of spatial coverage and not well considered through their dependencies on critical ecotones such as areas in between waterbodies and hillslopes or urban areas. Most butterfly data are centered around urban parks without detailed full coverage. This is in a broader eco-hydrological view [7,31], where water networks are everywhere under different water compartments (such as soil moisture in drier habitats) and under different water stress due to climate and water use. Therefore, butterflies are extremely sensitive indicators of both hydroclimatic variability (conditional to park structure) and associated species populations. As a result, butterflies are a fingerprint of ecosystem function (and vice versa, its dysbiosis) driven by balanced species–climate interactions structured by eco-hydrological proportions. At the macroscale, Brown and Freitas [18] (focusing on coastal forest butterflies in very natural conditions) found a clear signature of the environment in butterfly richness, such as landscape connectivity alone or by composite indices of environmental heterogeneity (disturbance, seasonality, temperature, vegetation, and soils), natural disturbance, and (negatively) anthropogenic disturbance. This differs from our approach, which created one multivariable non-linear index of suitability, which was later decomposed considering the non-linearity of composing variables. At the mesoscale, by using *Hymenoptera* as biodiversity indicators, the study of Kazemi et al. [19] examined how the implementation of water-sensitive urban design (WSUD) using bioretention swales in place of conventional urban greenspaces can influence urban biodiversity. The results of Kazemi et al. [19] show that even eco-engineering solutions (i.e., eco-hydrological solutions, to be precise) can positively influence butterflies. This can be achieved in bioretention swales by a higher number of plant taxa, higher coverage of mid-stratum vegetation, and optimal soil pH ranges, which are the main habitat factors attracting *Hymenopteras* to these landscapes. Brown et al. [17] also investigated how the habitat value of bee and butterfly species can be rapidly enhanced by converting standard road verges into native understory plants and that these benefits may be greatest for those most negatively impacted by urbanization. However, a high-resolution world-scale study is necessary to infer generalizable patterns of richness and abundance as a function of eco-hydrological changes in cities considering elements of the urban fabric at different scales. This is also considering that butterflies and other sensitive species are declining worldwide, including in places where urbanization is not as massive, spatially speaking, as in other countries such as Australia (see Braby et al. [16]).

As for species reporting, it is well known that biodiversity time series (of any species and not only butterflies) are biased towards increasing species richness in changing environments [32], where the largest changes have occurred in urban ecosystems in the last 30 years. This is related to density-dependent effects (more species are found in decreasing biodiversity areas that are more easily monitored, such as urban parks) or more reporting occurring whenever natural areas become closer to developed areas. The latter is, for instance, the case of ecotones close to pristine forests and cities. As a result, the absence of richness trends over time can actually reflect a negative deviation from the apparent positive biodiversity trend that is due to two diverging elements: increasing monitoring and decreasing natural habitats, where the balance between these aspects is geography-specific. Regardless of the limitation of data, species information, such as that of butterflies, is useful for a broad predictive purpose. *Pre*dictions (literally meaning “to mention beforehand” in Latin) are useful not only to assess future conditions of species but also for the inference of ecological corridors and preferential sinks, as well as the potential identification of species, such as plants supporting beneficial species interactions for the habitat where butterflies exist. This is particularly important for urban parks that are made with exotic species, which can potentially establish novel ecological niches for endemic butterflies.

### 1.3. A Structural Ecological Framework for Ecosystem Assessment and Design: The Ecosystem Fitness Index

Previous studies have explored various aspects of butterflies, including their ability to adapt to climate change [33], the impacts of urbanization on their populations [34], and how public perceptions of these species can inform conservation strategies [35,36].

However, conservation cannot just be about protecting one single species or habitat at a time; rather, a consequentialist approach to eco-hydrology as the core of ecosystemic function is needed. This can be pursued via the proposed structural ecological approach where eco-hydrological information on ecosystemic stress and likely risks affecting all species/habitats and cascading services is meaningful. In this purview we emphasize the necessity of ecosymbiosis, i.e., the balance between the natural and built environment jointly, achieved by a key ecological portfolio [1] under pressure as a function of climate impact and development in the future.

Some recent studies have focused on ecosystemic fitness by developing indicators that comprehensively assess the function and services of ecosystems. The SEED index [37] is a major index focused on the structure and function of ecosystems at multiple scales. Similarly, Zhang and Convertino [10] developed the ecosystem health index for blue carbon ecotones, where health accounts for ecosystem services these habitats provide beyond ecological stress. Hansen et al. [38] introduced the forest structural condition index (SCI) and the forest structural integrity index (FSII), considering plant traits. The forest landscape integrity index (FLII) was similarly developed to measure the condition of forests, considering environmental pressure in the form of proximity to urban infrastructure and agriculture [39]. One of the limitations of many of these indices is related to the lack of reference to a specific timeframe, whether the indices refer to the status quo, a set past, or future conditions under scenarios of change. The choice of a temporal reference should be the first step in deciding which indicators to use or analyze since different ecological features change with different velocities.

We provide a template for a model of eco-assessment and design based on a formulated ecosystem fitness index (EFI) that considers only ecological conditions. The index is based on the hydroclimatic habitat suitability (HS) (inferring ecological corridors and flows defining ecotones of ecosystems) and the divergence of relative species abundance (RSA) plots from the optimal Preston plot. This index ranks urban parks based on the community response to climate pressure and current species organization by merging two essential biodiversity variables or their post-processing (see [14]) into one systemic index of eco-fitness. The inference of ecological ties (corridors and flows) is based on assessing a multicriteria flow resistance (or habitat suitability and vice versa) whose minimum gradient defines the preferential directions and flows of species. Specifically, first, we use a MaxEnt model to predict the species’ HS, which is the opposite of the resistance surface in terms of the movement of species. The retained environmental features, predicting the butterfly climatic niche, are selected based on their non-linear functional interactions. From there, an eco-hydro-based inference of ecological ties (corridors and flows, their area of influence, and convergence as ecotones) is made considering the maximum HS gradient underpinning patterns of HS, with particular attention paid to invariant and extreme changes.

The inference model of ecological networks is based on a more generic minimum systemic entropy/transfer entropy (TE) model developed by Servadio and Convertino [27] (which was later expanded into a broader ecological context by Li and Convertino [3]) that explores how species interaction networks sculpt biodiversity for the microbiome [40]. Maximum HS gradients are those with the largest TE, representing the predictable extremes of HS patterns; these extremes converge (as information) toward the main pathways of ecological flows channeling collective behavior. Here, we propose a preliminary attempt to look into how these interactions play a role in shaping key species richness and abundance in large-scale urban ecosystems for their fitness assessment. Ecological flows are the fabric of species interactions mediated by their sensing of the environment, where the distribution of the latter defines how climate stress is distributed. Therefore, it is essential to consider eco-flows for engineering critical transitions via environmental determinants, given the information on interconnected *stress–response functions* of habitats (ecological communities) to climate pressure.

## 2. Materials and Methods: Digital Ecosystem Models

### 2.1. Data

The WorldClim data (https://www.worldclim.org/data/worldclim21.html (accessed on 16 April 2025)) used in this study have a spatial resolution of 2.5 min (approximately 4.625 km). These data refer to averages for the years 1970–2000, yet represent the average climate forcing of a recent historical past that does not differ much from the current average. To calculate abundance and richness (Table 1), we utilize data from the Shenzhen City Butterflies Project (Project Code: SCBP; https://v4.boldsystems.org/index.php/Public_SearchTerms (accessed on 16 April 2025)), which includes 1933 butterfly records collected from 10 parks. Local species richness, i.e., the number of species in a given area, is calculated in this paper instead of species diversity, which accounts for abundance. The latter serves as an indicator representing maximum diversity for species distributed evenly while presuming evenness to be the ideal condition for biodiversity. Species evenness, however, is not always the optimal condition for biodiversity. While high evenness, where all species have similar population sizes, can indicate a balanced ecosystem, many natural ecosystems thrive with uneven species distributions. Thus, the species-abundance pattern, reflecting ecosystemic fitness, is considered for Shenzhen butterflies.

### 2.2. MaxEnt and Multiplex Habitat Suitability Landscape

MaxEnt is used as a multiplexing model to predict the habitat suitability of butterflies considering a set of Worldclim environmental layers (Table 2). We apply the MaxEnt model (version 3.4.4) [41], a neuro-inspired model of information selection and inference, to predict the distribution of butterflies (habitats) and their relationships with environmental factors based on butterfly distribution data. This is the first time, to our knowledge, that MaxEnt has been used to predict habitat distributions (assumed as butterfly communities) rather than species distributions; some species-specific applications were realized in the past (e.g., see Chowdhury et al. [42]). Convertino et al. [43] and Convertino et al. [44] predicted habitat hydroclimatic flood and landslide risks in previous studies, respectively; however, they did not infer the underlying functional networks but looked into the contribution of hydrologic networks to risks. Ecological networks among species were previously inferred using maximum entropy approaches on relative species abundance data, such as the studies by Li and Convertino [3] for fish and by Galbraith et al. [40] for the ocean microbiome, and used with graph neural networks to forecast bioaquatic risks [9] such as algal blooms. In the context of Shenzhen, we recognize that the realistic butterfly distribution is very hard to obtain, and most of the data on butterfly occurrences are attributed to sampling transects across large areas of parks. In these areas, the microclimate can be relatively well defined; therefore, the microclimate scale defined by vegetated habitats should be the scale to which the suitability model refers. Furthermore, the niche of butterflies is defined by the microclimate imprinted by habitat features, and butterflies are a fingerprint of those (yet they are sentinel species of habitat changes). MaxEnt identifies the main environmental factors affecting the distribution of butterflies and predicts their potential distribution under any climate scenario. In doing so, MaxEnt begins with the ”neutral” hypothesis of fitting distributions as uniform distributions and updates the predictions by selecting those non-uniform distributions with the largest information power. The environmental covariates c are WorldClim variables for the current climate scenario, as listed in Table 2.

The Habitat suitability (HS) (note that here we adopt a macroecological characterization of the probability [44], i.e., the average suitability of a community to host butterflies) HS=P(y=1|c(t)) is calculated as follows:(1)$$HS_{i}=f\left(c\left(W_{ij}\right)\right)\;\exp^{\eta (c\left(W_{ij}\right))}\;\frac{P(y=1)}{f(c\left(W_{ij}\right))}\;,$$
where f(c) is the probability density function (pdf) of covariate c, and η(c)=α+ρh(c). α is a normalizing constant that ensures that $f_{1}\left(c\right)$ integrates to one ($f_{1}$ being the pdf of butterfly occurrences), and ρ is the constant (Lagrangian multiplier) of the MaxEnt features h(c) [45,46,47,48]. c can include any environmental layer as gridded information or digitized as networks $W_{ij}$, such as structural green and gray infrastructure networks. The Lagrangian multiplier that multiplies all environmental features determined the optimal trade-off between model complexity (defined as the number of environmental variables used as predictors) and model accuracy (that is, the distance between predictions and data) [44,46]. The set of parameters is identified by minimizing the prediction error between the observed and modeled floods. Features are transformations of covariates in the covariate space (that is, the multidimensional space of covariates), and this allows for faster and more precise computation rather than operating in the geographical space [44,45].

To reduce model complexity without compromising performance, we build several models by varying the feature classes (FCs, fitting covariates) and regularization multipliers (RMs) [49], using R 4.2.1 and the “ENMeval” version 0.3.0 package [50]. FCs determine the flexibility of the modeled response to predictor variables, while RMs penalize model complexity [48]. This model design is carried out to identify the most salient eco-environmental variables with the highest value of information for the predicted patterns [51] constrained on butterfly occurrence and the imposed feature classes that establish the distribution of suitability around occurrences. We randomly divide occurrence records by 70% for the model selection (i.e., feature selection). ENMeval partitions the localities internally to test each combination of settings, so we use the random k-fold method to divide localities into four bins. We build models with three FC combinations (LQ, LH, and LQHP, where features are linear (L), quadratic (Q), hinge (H), and product (P)) and vary regularization multiplier (RM) values from 1.0 to 5.0 in increments of 1.0. The optimal models are selected using Akaike’s information criterion, which has been corrected for small sample sizes (ΔAICc = 0); this approach penalizes excessive model complexity and facilitates the selection of models with an optimal number of parameters [52]. The area under the curve (AUC) refers to the area under the receiver operating characteristic (ROC) curve, which is a standard metric used to evaluate the performance of a model by measuring its ability to distinguish between positive and negative predictions, with a higher AUC indicating a better model fit. The AUC of a random prediction on randomly distributed background points is 0.5 because an equal probability is used for predicting both positives and negatives. Therefore, it should be noted that the predictions’ accuracy of MaxEnt is measured against random prediction [44,53].

### 2.3. Digital Ecosystem Models: Inference of Ecological Flows and Attraction Basins

From an eco-hydrological perspective that aims to link critical species–hydroclimate dynamics (at least in a predictive sense), **preferential flow directions** are defined by the steepest gradient of HS, which is the minimum resistance in the movement of species for any pixel. The model of eco-tie extraction is based on a more generic transfer entropy model (TE) developed by Li and Convertino [3] and later expanded by Wang and Convertino [9] for space–time forecasts. Maximum HS gradients are those with the largest TE because they have the largest divergence in habitat suitability along and across the pathways in HS patterns. These pathways of HS converge as information to form ecological flows channeling the collective behavior of the vast majority of species. Maximum gradient pathways that are preferential pathways are defined as follows:(2)$$EN_{g}=\arg\;\max\left[\frac{\left(HS_{i}-HS_{j}\right)}{\delta x_{ij}}\right]\;,$$
where $\delta x_{ij}$ is the pixel-defined distance, i.e., five arc minutes (about 9.2 km at the equator). The maximum gradient in HS, over HS as a digital ecosystem model (DEM), defines the minimum resistance pathways where species moves with the minimum metabolic consumption [54]. This is an analogy of water flows extracted from digital elevation models [55]. These eco-flows, defined everywhere, are, in principle, bidirectional with loops; however, the flows of species is likely to be from low to high HS areas due to the search for higher-quality habitats, with weak looping for the minimum energy principle [56]. The **cumulated ecological flow**, which similar to the cumulated drainage area in hydrogeomorphology, is defined by the sum of all links upstream of a point as follows:(3)$$E_{i}=\sum_{j}A_{ij}\left(EN_{g}\right)\cdot E_{j}\;,$$
where $E_{j}$ are upstream ecological fluxes (i.e., fluxes of information, matter, and/or energy, such as species abundance that also depends on structural green infrastructure corridors; see Brose et al. [20]) beginning from $E_{j}=\delta HS_{ij}/\delta x_{ij}$ for source fluxes (i.e., one unit of species) and increasing downstream following the maximum gradient $EN_{g}$ in Equation (Equation 2). $A_{ij}=1$ is the adjacency matrix (or *morphological systemic “epigraph”* as the graph responsible for the ecosystemic flow distribution that is also blueprinted by the underlying structural networks $W_{ij}$ determining HS in Equation (Equation 1)) for communities/pixels that belong to OEN; otherwise, $A_{ij}=0$. Here, we consider the top 80 and 20% (lower and upper thresholds) of ecological flows determined by Equation (Equation 3) representing uncommon and preferential/extreme flows. Functional **optimal ecological networks** (OENs), which are the most predictive information networks of observed ecological patterns (see Li and Convertino [3]) considering data uncertainty and systemic information, can be defined as thresholded maximum gradient pathways as follows:(4)$$OEN\;\;\;\mathrm{for}\;\;\;E_{i}\geq E_{th}\;,$$
where $E_{th}$ is the threshold of significance for common and rare species flow. Further research is required to fine-tune this threshold in relation to minimum water flows supporting connected ecological communities forming ecosystems. In an eco-hydrological analogy, the threshold is equivalent to the one that separates channelized and unchannelized (hillslope) flows in river basins. Note that here, fluxes can be calculated considering both directions, upstream and downstream, along the maximum gradient pathways due to the possibility of species moving in both directions but more likely from low to high suitability habitat, thereby establishing a de facto movement bias, where ecological sinks or attraction basins (like urban parks) act as attractors. The meaning of cumulated flow reflects the total relative magnitude of species flow in one location across the ecosystem considered. The precise information of RSA over time can be used by transfer entropy models to quantify ecological flows that are not anchored into the maximum gradient assumption but respond to higher non-linear dynamics, as in Li and Convertino [3].

The detection of **ecological basin boundaries** (defining basins or sinks across basin divides that attract species to similar climate variables) related to ecological flows are defined by the maximum divergence between flows diverging in opposite directions (in two or more basins) independently of the magnitude of the gradient of HS:(5)$$D_{KL}=\sum_{i,k}p\left(E_{i}\right)\;\log\;\frac{p(E_{i})}{p(E_{k})}\;,$$
where $D_{KL}$ is the Kullback–Leibler divergence over $E_{i}$ and $E_{ik}$, which are the flows defined by Equation (Equation 3) with opposite directions, yet not belonging to the same networks. Equivalently, in a simpler geometrical definition, basin divides are defined by the curvature of the HS (or first derivative of gradient EN in Equation (Equation 2)), which is positive:(6)$$EN_{c}=\frac{\delta^{2}HS_{ij}}{\delta^{2}x_{ij}}>0\;.$$

The curvature of a drainage ridge is strictly positive, meaning that flow diverges at the ridge into two different basins. Roughly speaking, the convex zones are hillslope-like zones. Instead, the curvature of a valley is negative, meaning that flow converges at the valley and eventually forms flows. These eco-basins are equivalent to sub-basins in hydrology and potentially compartmentalize flows into independent areas (eco-sinks). Eco-basin boundaries are then equivalent to drainage divides in river basins.

**Local ecological sinks** are points where the HS gradient is at its maximum (as in Equation (Equation 2), they are not necessarily points with the highest HS), and yet flows converge to these points. Therefore, ecological sinks are points along the ecological network (above a threshold on flow, as shown in Equation (Equation 4)). **Global ecological sinks** are points with the highest HS gradient and HS. The outlet of networks (with the largest cumulated flow, as shown in Equation (Equation 3)) is often not the point with the largest HS or HS gradient. Analytically, global ecological sinks are defined by the simultaneous maximization of two functions:(7)$$ES=\left\{\begin{array}{c} \max\;(EN_{g})\\ \max\;(HS)\;. \end{array}\right.$$

From a network perspective, sinks are characterized as links (along shortest-path trees on the networks that measure their systemic influence) via the use of the ecological diameter. The **ecological diameter**, $\Phi_{i}$, is defined as the average distance from stream link i to all other stream links j in the network, where the distance is measured through the network defined by ecological flows. The link with the smallest diameter in the network (the key salient link with the highest link centrality, as shown in Galbraith et al. [40]), i.e., the link with the shortest path to all others (approximatively at the center of the network; see Convertino et al. [57]), is a global ecological sink. On average, it is characterized by the conditions of Equation (Equation 7). Analytically, it is formulated as follows:(8)$$\Phi_{i}=\min\frac{\sum_{i,j}d_{i,j}\left(EN_{g}\right)}{N}\;,$$
where $d_{i,j}\left(EN_{g}\right)$ are the geometrical distances considering ecological fluxes, and N is the total number of links. Then, $\Phi_{i}$ is a measure of the degree of connectivity associated with a link, and it is crucial for the dispersal dynamics of species given HS. Distances $d_{i,j}\left(EN_{g}\right)$ are also used to assess the effective interconnectedness of urban parks and their proximity to the mapped forest landscape integrity index (FLII) [39].

### 2.4. Ecosystem Fitness Index

The ecosystem fitness index (EFI) was introduced as a systemic index, an analogy of the ecosystem health index proposed by Zhang and Convertino [10] (this includes ecosystem services, and is, therefore, a more consequential index), to capture the combined systemic response capacity of sentinel species (representing communities) to hydroclimatic pressure (as stress-response functions) and the current community richness divergence from the optimality condition (i.e., the log-normal Preston plot of the relative species abundance dependent on the optimal organization of habitats, which is also the most predictive because it is non-randomly organized). The latter, which reflects the current ecological conditions of species and habitats, including bio-spatial interaction effects [3], is calculated as the divergence (Kullback–Leibler divergence, or KL) of the observed relative species abundance distribution ($RSA_{o}$) from the optimal log-normal Preston’s relative species abundance distribution ($RSA_{P}$). The EFI, which considers only ecological conditions, is analytically defined as follows:(9)$$EFI_{p}=\underset{\mathrm{eco}-\mathrm{climatic}\;\mathrm{fitness}}{\underset{︸}{\langle HS\rangle }}+\underset{\mathrm{community}\;\mathrm{fitness}}{\underset{︸}{\left(1-KL(RSA_{o},RSA_{P}\right)}}\;.$$

In this context, the RSA is only for butterfly species, but for a systemic analysis, all species should be potentially considered. Other species beyond butterflies can be investigated as optimal eco-indicators of parks’ response to climate forcing. The EFI was calculated for urban parks to rank them in terms of their systemic eco-fitness defined by their ecological condition that is dependent on local species fitness (via HS, where species are eco-indicators) and connectivity [56] (likely manifested by the RSA that depends on species dispersal [11]). The EFI is normalized in a [0,1] range as a constructed multivariate index considering the maximum $EFI_{p}$. Then, the EFI is discussed as a function of the forest landscape integrity index [39] and the functional interconnectedness defined by the inferred ecological network (Equation (Equation 4)) and green infrastructural corridors.

## 3. Results

In Figure 1, we show the main ideas about how eco-information, in this case of butterflies, can act as a barometer of ecosystem fitness such as of urban parks, and then address ecosystem diagnostics, design, and eco-engineering. Figure 1B shows the preferential pathways over the digital habitat suitability model (DHSM) (i.e., maximum gradients, as defined in Equation (Equation 2)) representing the convolution of compounding hydroclimatic features onto species occurrences. Preferential flows (Equation (Equation 3)) are those with the highest predictability of HS across ecosystems, yet they are the most important features species are sensitive to when moving in the landscape. The opposite of HS is the systemic environmental pressure or resistance that minimizes the movement and fitness of species along resistance corridors. The probability density functions (pdfs) of environmental pressure and eco-flows are shown in corresponding colors to represent eco-corridors: the larger the average environmental pressure, the lower the eco-flow and vice versa (in black and red for low and high eco-flows, respectively). The principle of minimum resistance or maximum fitness has been shown to hold in any ecosystem at different scales [56] and is proposed here in the context of urban ecosystem assessment by leveraging butterfly data. Any other data can certainly be included in the model. The attraction basin of the network (or ecological basin) is identified by the positive curvature of HS areas and delineated with dashed lines in Figure 1B. This framework, an analogy of river basins, can be used to define and assess ecological stress as a function of HS, gradient, velocity, and acceleration (i.e., the rate of change in HS over space–time).

The predicted HS and ecological flows are shown in Figure 2A. The spatial predictability of HS is very low, as manifested by the limited heterogeneity of HS over space. This is largely due to the extremely limited data on space concentrated in urban parks placed in the historical core of the city (Nanshan, Futian, and Luohu districts from west to east). The very localized distribution of HS is also related to the large monitoring bias that is highly concentrated in the center of the city, where a variety of parks have been built with a relatively high exotic plant diversity; the latter constitutes an attraction for diverse butterflies. Another high HS area is in the East Dapeng area (shown as red pixels on the eastern-most side of Shenzhen along the coastline), which is a well-preserved natural area with very little development. HS ended up being quite high, although we did not have any butterfly data for this part of the city. The ecological flows in Figure 2B, calculated as in Equation (Equation 4), define five preferential pathways (with top 20% cumulated flows) that are relevant for butterflies and potentially other species, from low to high HS areas and pointing to major parks as attractors. The size of the cumulated flows is a measure of the importance/attractiveness of parks, and it shows the speed of attractiveness based on the magnitude and rate of change in the HS gradient toward a point (that is, the second derivative of HS over space). The purple points in Figure 2B are connection points or final points of convergence of the preferential flows, where the direction travels from low to high HS along maximum gradients. The convergence of some of these purple points toward parks is a good indicator that parks act as connected ecological attractors; furthermore, the lack of convergence indicates a missing park or a corridor in the urban landscape, considering the potential suitability driven by climate features. The red, yellow, and green dashed lines are *ecoclines* reflecting decreasing HS gradients and HS, as well as proximity to high FLII and EFI; the red ecoclines delineate systemic ecological sinks (with the highest EFI) such as Tanglangshan, Meilin, Donghu, and Linahuashan parks.

Attraction basins of species (eco-sinks) shown in Figure 2C, defined by Equation (Equation 5), are independent areas with similar eco-climatic niches, including potential species movement. Inside each basin, flows converge and are independent as one network, reflecting the accumulation of flows and divergence from other flows. The definition of these basins is really important for targeting areas of control and restoration that are aware of species movement. Figure 2C also shows that the clustering of eco-flows (black lines) is quite high for a variety of standing water bodies (reservoirs and lakes), but few flows coincide with the hydrologic network (streams and rivers of different orders). Interestingly, for Shenzhen, the eco-flow network has much more extended scale-free connectivity and clustering for low HS gradient areas (green areas), populated by large reservoirs that act as sources of ecological corridors “draining” into urban parks; the latter have higher HS and HS gradients and small-world connectivity that does not overlap with the hydrologic network. This may signify an ecological opportunity to connect these areas via vegetated ecological corridors along water courses in the city, an effort that is occurring in the city of Shenzhen [58,59]. In Figure 2C, it is also possible to note that the hydrologic network does not overlap much with urban parks, and that is a suboptimal condition (due to the low presence of water in parks) and a potential cause for the anomaly of patterns like the Preston plot in centrally located parks. Ecological dynamics cannot properly exist without vital hydrological dynamics; therefore, it is ideal to have a substantial overlap of eco- and hydro-flows supporting fitted communities [55].

Figure 3 shows the ecological response functions that are the HS probability conditionals to hydroclimatological variables when considered in isolation (red line). The blue shades consider the variability of the HS probability when all variables are varying simultaneously across the domain of analysis. We show the top five most important hydroclimatological variables whose non-linear interactions (as permutation importance in Table 2) predict the largest variability in HS. These variables are the most predictive and likely the most causal for butterflies’ variability. The response curve values shaded in red (see the x-axes on the response curves) are the tipping point of HS for the top-five environmental determinants. Those values lead to the largest changes in HS, where all transitions are of the second order, meaning they are gradual and not sudden. The critical thresholds were identified only for hydroclimatic variables and not for other urban ecosystems’s features. These factors do not act in isolation, so the systemic critical threshold of butterfly HS is the multiplicative composition of these thresholds, as in Servadio and Convertino [27]. The most important variables according to the permutation importance (Table 2), i.e., the mean temperature of the coldest quarter (BIO11) and the precipitation of the coldest quarter (BIO19), increase and decrease HS gradually but quickly and determine the largest change in probability for HS. The importance of these climate variables is a reflection that butterflies are good sentinels of evapotranspiration, as hypothesized. The associated response curves help to identify critical tipping points of temperature and precipitation that lead to major critical transitions, such as HS=0. Further research will explore how these optimal temperature and precipitation ranges for the ecological communities can be constructed by the design and engineering of critical urban ecosystem features.

The permutation importance of MaxEnt (Table 2) is based on directly disrupting the relationship between a feature (hydro-climatological variables) and the target variable (HS) by randomly shuffling its values and keeping all other features as changing. Permutation importance is a measure of the total non-linear interactions of features [51]. The percentage contribution keeps all other variables fixed and can be influenced by the optimization path taken during model training, making it potentially less consistent across different runs. On the contrary, the importance of permutation depends only on the final MaxEnt model, which is the one with the highest predictability considering the whole variability, not the path used to obtain it. Thus, the permutation importance was and must be considered when ranking the importance of variables, such as the ones leading the HS of butterflies’ habitats.

The contribution of each variable is determined by randomly permuting the values of that variable among the training points (both presence and background, where the background points serve to represent the random species prediction) and measuring the resulting decrease in training AUC. A large decrease indicates that the model depends heavily on that variable. Values are normalized to provide percentages of the variables’ contribution. The percentage contribution depends on the path of the MaxEnt model, and for highly correlated environmental variables, the percentage contribution should be interpreted with caution. In our study, in order of importance (considering permutation importance as high-order interactions), the key hydroclimatic factors in shaping HS are as follows: BIO11 = mean temperature of coldest quarter; BIO19 = precipitation of the coldest quarter; BIO4 = temperature seasonality (standard deviation ×100); BIO5 = maximum temperature of the warmest month; BIO10 = mean temperature of the warmest quarter.

The relationships between abundance, diversity, and HS for our study area and data are shown in Figure 4. Both abundance and diversity (Table 1) are mildly associated with park area according to a power function, unrelated to longitude but related to the degree of urbanization/park centrality, counterintuitively. This is possibly associated with the isolation of high-richness parks carrying high endemic diversity or the elevated introduction of exotic species. These inferred patterns also show trends that are not commonly found in natural ecosystems, where the species–area and abundance–area relationships increase non-linear functions. Additionally, we also found the following: (i) a U-shape abundance–diversity pattern reflecting species heterogeneity and rarity of low and high diversity and (ii) abundance and diversity increasing and decreasing as a power-law function with HS. The generalizability and stability of these patterns require further studies considering butterfly data and urban ecosystems worldwide.

Figure 5 and Figure 6 show the species–abundance Preston plots and the abundance–rank relationship for each park in Shenzhen. Ecological patterns such as the Preston plot and abundance rank can reflect the stationarity of ecological communities when they are log-normally or exponentially distributed according to the theoretical expectation reflecting an ideal log-normal distribution of the relative species abundance. However, the optimality of these patterns cannot be informative of the level of endemicity of parks, a park’s proximity to change, and their relative connectivity; all of these elements should be addressed considering temporal data and key species data. Regardless, based on the available species data, we ranked urban parks based on the EFI (Section 2.4) as a function of HS and Preston’s optimality from high to low as follows:(A) Tanglangshan, Meilin, Donghu, and Linahuashan are healthy parks with the highest joint fitness (as HS and convergence to the log-normal Preston plot). These parks also have a relatively high interconnectedness with each other and are not directly proximal to areas with high residential density. However, they have a high FLII (as in Grantham et al. [39] available at https://www.forestintegrity.com/ (accessed on 16 April 2025)) or are proximal to forests with high FLIIs and hydrogeomorphic heterogeneity (Tanglangshan has the only forest overlapping with the whole park in West–Central Shenzhen). These high-EFI sites are ecotones between natural areas and densely populated areas.(B) Lithchi, SZ University, and Honghu have intermediate fitness due to the higher percentage of high-abundance classes that make the Preston plot better, even across species. These parks are much smaller and closer to areas with high residential density.(C) Huanggaong, SZ Central, and SZ Bay leisure greenway have a low EFI because of their bimodal Preston plots, indicating two bistable states in species–abundance. This bistability may suggest some instability compared to other parks. These parks are smaller and closer to high-residential-density areas. These areas are much further from high-FLII areas and have very low hydrogeomorphic variability.

## 4. Discussion

The decision protocol for ecosystem fitness assessment was synthesized and is shown in Figure 1. Biodiversity information, only provided as species richness, has limited value and is not actionable for vast ecosystem decision-making because it is not linked to any systemic function or response, such as climate. The value of information on biodiversity (as species richness) is inconsequential. One of the major contributions of this paper is providing precise values and ecological information and creating a modeling framework for actionable information for ecosystem assessment and decision-making (strategies, such as restoration, enhancement, and creation). This is based on leveraging eco-environmental data to calculate HS as the systemic fitness function in response to climate and RSA divergence from optimal patterns to the EFI as a proxy of community stability and systemic function.

According to Clapeyron’s law, a one-degree Celsius increase in temperature results in approximately a 7% increase in the amount of evapotranspiration (ET) (water vapor) the air can hold. Therefore, even just an increase of 1.5 degrees Celsius could result in ∼9% more water in the atmosphere, which could have a major impact on local storm systems, subsequent rainfall, and potential flash floods and droughts (in the short- and long-term). That is why the restoration of vegetated ecotones is critical not only for butterflies but for hydroclimatic regulation, considering the feedback between surface and vertical water fluxes. In this perspective, butterflies can be excellent sentinels of change, such as for ET, considering their sensitivity to these invisible yet important fluxes. Thus, butterflies are informative of very local conditions and can be easily observed in contrast to ET, which requires specific sensors.

### 4.1. Modeling Innovations: Inference of the Functional Ecological Architecture

In physics, butterfly phenomena are known as those where small shifts in one variable (like temperature, which is seemingly uninformative) can lead to an anomalous or extreme change in another variable (e.g., butterflies, which are informative). In this framework, we hypothesized (supported by evidence) that butterflies change much more than underlying environmental determinants, such as climate and park feature variability. Butterflies act as a fingerprint of park fitness in response to climate pressure due to their larger information variability that leads to a high *value of information* [60]. From a *reverse causality* perspective, butterflies are very sensitive species to explore the future impact of climate extremes (such as temperature-driven evapotranspiration) on ecological communities in urban ecosystems.

In this causality inference, *eco-hydrological networks and flows*, as the foundational ecological architecture, are important for ecosystem fitness assessment and design because they highlight the preferential corridors of change and their magnitude. Networks and flows constitute the *eco-connectome* [9], which has physical meaning in a biological sense [20] and is salient for predictions and associated decisions (thus, sculpting the “architecture of decisions”). In this paper, we introduced three main features that support this causal inference and can be integrated into digital ecosystem models or digital ecological models (DEM), where the latter contains only ecological information useful for ecoclimate-informed decision-making. Specifically, DHSMs are “ecological landscapes” for species like digital elevation models are for water, and extracted eco-flows are like water flows. The case study was conducted for Shenzhen, which is the fastest-growing megalopolis in the world. The three main features of DHSM, which are related to the hydro-physics of ecological processes (explaining the model structures in physical elements), are as follows:**Eco-functional networks and flows**: From an eco-hydrological perspective, preferential eco-flow networks are defined by the connected steepest gradient paths of habitat suitability (HS) (Equation (Equation 2)) that offer the minimum resistance, approximating the flows of species across the landscape [61]. The cumulated flow, similar to the cumulated drainage area in hydrogeomorphology, approximates the potential cumulated abundance of species living on the landscape and moving along the preferential network [7,55]. This potential cumulated abundance provides a sense of the relative proportion of species in a landscape, considering all available preferential pathways.**Eco-attraction basins**: Eco-basins, equivalent to hydrological basins, are landscape areas where ecological flows converge into one network that is independent of others and reaches an ecological sink or outlet. Eco-basin boundaries (defining attraction basins and sinks that attract species from low to high climate-based HS) are defined by the maximum divergence between flows directed toward opposite directions (for one or more adjacent basins; see Equations (Equation 5) and (Equation 6)), independently of the magnitude of HS gradients. Eco-basin boundaries are equivalent to drainage divides in river basins. In a simpler geometrical definition, basin divides are defined by the curvature of HS that is greater than zero.**Ecological sinks**: Local ecological sinks are points where the HS gradient is the maximum (not necessarily points with the highest HS), and flows converge to these points. Therefore, ecological sinks are points along the ecological network (above a meaningful threshold on flows). Global eco-sinks are points with the highest HS gradient and HS (Equation (Equation 7)) toward where the vast majority of the flows converge. The outlet is often not the point with the largest HS. Often, the *network diameter*, which is the link with the shortest path to all other links (approximatively at the center of the network; see Convertino et al. [57]), is the global ecological sink.

These aforementioned features, and particularly ecological flows, were assessed over space but can be calculated over time at different temporal scales and resolutions if data are available and whether they are meaningful. For instance, for butterflies (and or other key sentinel species), it would be meaningful to calculate these quantities at the seasonal- or year-scale since environmental changes rarely occur on smaller temporal scales. For evolutionary studies, the scales should be much larger, like 5 to 20 or more years (also considering the adaptation capacity of species), while for assessing the butterfly response to extreme hydroclimatological events such as hurricanes, the scale should be much smaller (like days or weeks at most). Specifically, the consideration of ecological networks reflecting the collective dynamics of key species is a non-negligible factor for assessing community function in relation to the physical environment. In particular, the accounting of hydrosphere-related ecological flows allows us to define how much shifts of water flows (in its multifaceted aspects) affect species, communities, and climate in an interconnected way as feedback or “ecobonds” (species interactions, habitat interactions, and land–climate teleconnections) forming an ecological portfolio [9,61]. Given this information, a proactive “bioterraformation” approach, that is, eco-hydro-geomorphological engineering, should be devised and planned accurately.

### 4.2. Ecological Patterns and Systemic Fitness Indicators: Species as Hydroclimate Sentinels

Eco-fitness is the opposite of eco-stress and relates to the quality of biodiversity (ecological composition/morphology) vs. its quantity, such as species fitness, which is refers to collective species function under environmental conditions vs. metrics, such as species count. Eco-fitness can be measured by the convolution of the environment and key ecological functions, such as HS [61] or species interaction network features (i.e., the “eco-connectome” [9]) related to the RSA [8]. Both of these functions are combined into the EFI here (Equation (Equation 9)). This approach moves the discussion toward *ecological quality*, where eco-stress, referring to the change in systemic function, can trigger slow or catastrophic ecosystem collapse if not monitored. Thus, the inference of *functional ecological corridors* (Equation (Equation 4)), anchored into the water wiring of ecosystems [7], is informative of ecosystem fitness and shifts (i.e., flow change as eco-stress), critical monitoring areas, and improvement strategies to enhance ecological function. Further studies will look into how eco-stress, driven by eco-hydrological disorganization, leads to the generation of information about human risks such as heatwaves. This is in a broader ecosystem health perspective [1,62,63], where the consequential nature–human nexus is considered and engineered.

Specifically, HS downscales climate information to the landscape where species, as discrete occurrences, are transformation functions of continuous climate layers. Then, the HS model, as a digital ecosystem model, provides distributed maps of the species’ response to climate information that define suitable and unsuitable bioclimatic niches, as well as flows that are informative of potential routes of dispersal. Ranges of HS gradients identify areas of common dynamics, such as the red and green high and low HS gradient in Figure 2B. The former coincides with high HS areas where parks are located, manifesting high attractiveness for species and confined existence of suitable communities. Outside of these areas, much lower suitability exists with likely minimum climate-driven flow. Temperature and precipitation of the coldest quarter and seasonality were the most important variables in this study, underpinning the importance of critical features of the hydrologic cycle and its seasonal regularity. All these atmospherical features, routed through hydrological corridors and flows on landscapes, change dramatically across the globe, thus compromising the habitats and key species they support. Further research is necessary to test the generalizability of these findings on a global scale, including the stability of critical temperature and precipitation tipping points for ecological communities.

As for the linkages between HS and macroecological indicators (which are traditionally used to assess biodiversity), we found the following: (i) a U-shape abundance–richness pattern reflecting non-random species heterogeneity and rarity of high diversity (Figure 4), and (ii) abundance and richness increasing and decreasing in accordance with power-law functions with HS (Figure 4). Species richness appears to be related to park area and abundance more than the degree of urbanization or park centrality due to the likely isolation of species parks with higher species reporting. At the same time, central parks are characterized by bimodal Preston plots (species richness–abundance, or “RSA” as relative species abundance patterns; see Figure 5) that underlie some non-stationary and/or suboptimal conditions in species assemblage. Parks are then ranked considering HS and RSA divergence from optimality to compose the ecosystem fitness index (Equation (Equation 9)). It should be noted that the relationship between species richness and area is not reliant on the species–area relationship [64] because it does not refer to sampling species from an increasing sampling area within the same ecosystem; rather, the relationship is built considering the total richness in parks and their wider area.

### 4.3. Limitations and Perspectives

The main limitation is that the current study did not investigate whether other species can be better sentinels or complement the sentience of butterflies to eco-hydroclimatic changes. Birds, for instance, can be good eco-indicators of the fitness of habitats in parks. However, migrating birds also respond to long-range dependencies (natural areas, the built environment, including their vertical dimensions, and climate teleconnections) that are much larger than on the city scale, informing us of other large ecological impacts concerning the cities considered. Conversely, birds and other long-range species might be informative of how cities impact the broader ecology compared to the city alone, e.g., how much urban parks may affect the ecology and climate on a larger scale (such as positively affecting migration routes and population abundance and/or evaporative fluxes in other far-reaching places).

Regarding ecological flows, further assessment should be carried out to calibrate how the inferred HS flows to real species dispersal, which is certainly a species-specific calibration in terms of dispersal ranges. Nonetheless, HS gradients are suitable for mapping the preferential directions of dispersal that are missing from species distribution models despite those corridors being assumed by many ecological models, such as metacommunity models [7]. Testing the correspondence of eco-basins with hydro-basins and the inference of minimum water flows for vital ecological communities is also an important investigation that needs to be carried out for integrated water-centric ecosystem management.

As for the EFI, it is not meant to measure absolute fitness but relative fitness, considering elements within the same ecosystem and the indicators considered, i.e., hydroclimatic factors. Further testing is required to determine whether other patterns are crucial to predict community fitness beyond Preston’s plot and their relationship with the habitat’s structure, such as connectivity and flows. This is aligned with the mission of essential biodiversity variables [13,14,15] (EBVs) that was also introduced to advance the synthesis of biodiversity information. More investigations are needed to understand the robustness of the EFI and other ecological patterns under more systemic eco-hydrological variability of urban ecosystems (beyond temperature and precipitation) that also accounts for freshwater flows and multivariable pollution.

Lastly, strategic protocols for policymakers, urban planners, water managers, and designers should be devised regarding how to incorporate ecological stress-response functions and the EFI into ecosystem restoration and design that include climate and development plans. In this process, other sentinel species beyond butterflies and environmental layers can be considered to be informative of eco-hydroclimatic fitness. Further research will clarify the universality and specificity of the eco-environmental information in defining the EFI.

## 5. Conclusions

The analysis of species distribution and richness in urban ecosystems (i.e., cities considering natural and man-made assets simultaneously) should be a top priority for ecosystem fitness assessment and eco-informed decision-making about ecosystem management, restoration, and construction. The last three activities can be synthesized as *ecosystem enhancement* [1,4,5], where the salient systemic fitness is increased via the improvement of species–habitats–climate teleconnections [20,24] through hydrological corridors and flows (i.e., the effective morphology of eco-indication and fitness). This study provides an open, flexible, and explainable computational tool (and related indicators) vs. closed-form physical-based models for eco-fitness assessment and ecosystem design. This is particularly important for adaptive planning for the future under extreme climatic and development changes.

In this study, we assessed how much HS affects community features such as relative species abundance (RSA) and richness of butterflies and whether those variables, combined in a novel ecosystem fitness index, can be used as a potential barometer of ecosystem fitness. An important element related to species richness (i.e., species count) should not be used as a systemic indicator of ecosystem health or the conservation of natural ecosystem conditions. This is clearly observable in Shenzhen, where the highest richness and abundance, considering butterfly data, are observed for urban parks that are quite central in the city; these parks have the largest urbanization pressure considering the urban evolution of the city, and only a few of them with a large area and interconnections have a high EFI. Some of these parks are in the oldest parts of the city. Thus, other elements, such as the presence of exotic plants or ecological isolation, may contribute to these high values of diversity that do not necessarily reflect a high EFI. Thus, ecosystem fitness should be based on the topology of species interaction networks [3] from relative species abundance that supports community function, which can be easily assessed over space and time and should be structured along ecological corridors [7,55,61].

Another variable that is salient for baselining ecosystem fitness is the HS of species indicators, like butterflies, which are sensitive to and sentinels of, in the sense of vulnerability and early-warning signals, the hydroclimatic footprint and interdependent species. Specifically, HS flows (of optimal ecological networks manifesting function) are sentinels of changes due to their strong dependencies with temperature and precipitation shifts along hydrological corridors connecting the land and atmosphere. Eco-flows incorporate species and species-transformed climate pressure into a distribution reflecting compounding and distributed stress response. Thus, eco-flow distribution, more than RSA by itself, is a sentinel of change that manifests the balance between climate forcing and species response. This perspective can be seen in an information sense where ecological information becomes intelligence for decision-making on ecosystem management and design [1,3,9]. Ecological information can support digital ecosystem models (DEM) that provide community fitness (proposed here as HS of sentinel species), ecological corridors and flows, areas of influence as ecological basins, and environmental determinants to manage and/or reshape (such as park configuration and connectivity). DEMs are flexible to include any information deemed relevant to decision-making, such as detailed water stress information [31,65], high-resolution land-cover/land-use [66], and the forest landscape integrity index [39].

In the context of rapid ecosystem change, ecotones of eco-hydrological corridors (as *vegetated corridors and water flows*, where ecotones are more broadly critical connectors of two or more distinct ecosystems) defined by the proposed model should be adopted as critical elements in shaping the ecosystemic function and condition of key species populations and ecological communities. Thus, defined ecological flows based on MaxEnt maximum gradients (of indicator species related to endemic and exotic vegetation, such as butterflies) can help the design of more ecofunctional cities for desired climate regimes that regulate HS. In a city, the presence of urban parks alone is not enough because these parks must be connected together, and the details of ecotones (such as which plant species is introduced) are important in shaping the ecological architecture defining the systemic function (and fitness as a divergence from the optimal function). Tracking species allows stakeholders to determine the signature of the hydrologic cycle that may be shifting and to diagnose and plan ecosystems more properly than in a non-eco-informed scenario. Butterflies and other critically sensitive species act as fingerprints of ecosystem function (and vice versa, its dysbiosis) driven by the balanced species–climate interactions structured by eco-hydrological proportions.

## Figures and Tables

**Figure 1 entropy-27-00486-f001:**
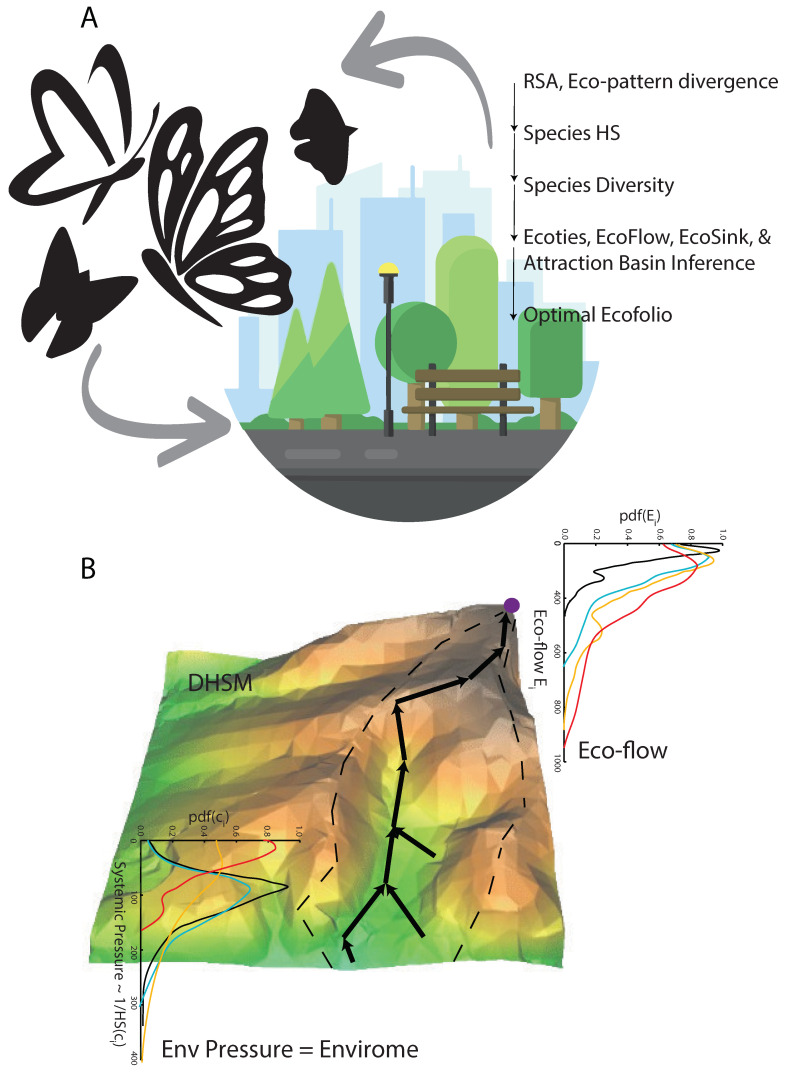
**Conceptual diagram of the inferred butterfly-based habitat suitability and flows for urban ecosystems.** (**A**) Butterflies are a potential fingerprint of park fitness (i.e., how much butterflies predict the EFI through HS and RSA, as shown in Equation (Equation 9)) considering hydroclimatic extremes that can impact ecological communities (i.e., the prediction of butterfly change due to hydroclimatic pressure from a predictive causality perspective). The physical hypothesis is that the higher the HS, the higher the ecological community’s fitness, and that butterflies are “harbingers” of larger climate impacts. The ecosystem diagnostic framework of the EFI is shown on the right. (**B**) Graphical representation of the digital HS (digital habitat suitability model, DHSM) and inferred preferential flows (as maximum gradients), which are useful for ecosystem diagnostics and design.

**Figure 2 entropy-27-00486-f002:**
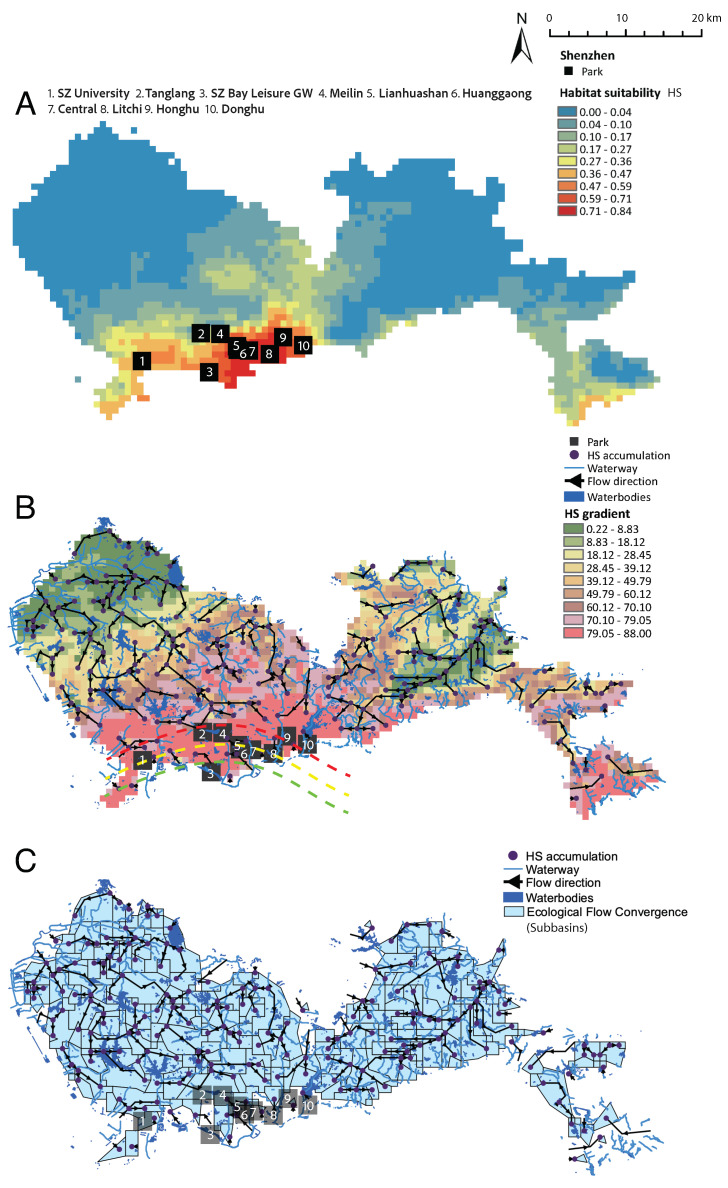
**Butterfly habitat suitability, maximum gradient flows, and attraction basins.** (**A**) Habitat suitability. (**B**) Optimal ecological networks as the steepest gradients of HS after a threshold on the minimum flow. The red, yellow, and green dashed lines are ecoclines reflecting decreasing HS gradient and HS as well as proximity to a high EFI; the red ecoclines delineate systemic ecological sinks (with the highest EFI) such as Tanglangshan, Meilin, Donghu, and Linahuashan parks. (**C**) Attraction basins (eco-sinks) of converging flows with similar eco-climatic niches, including potential species movement. Purple nodes are points where multiple connections exist, including endpoints. Sub-basins are nested areas within the whole ecosystems where one or more flows are jointly draining into one common point, and no other flow can enter the sub-basin.

**Figure 3 entropy-27-00486-f003:**
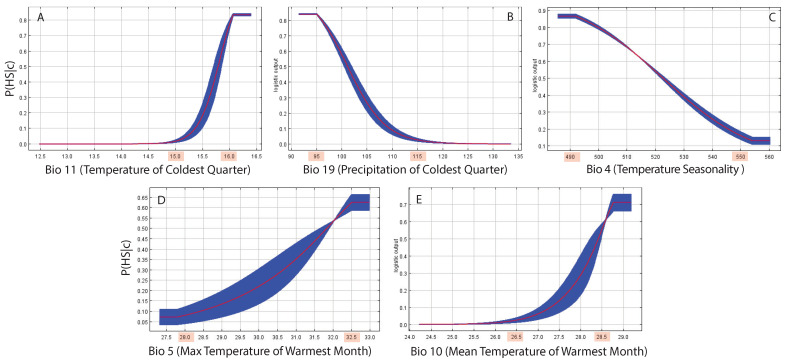
**Response curves as environment–ecology bivariate patterns for the top-five climatic features in terms of HS predictability.** The response curves are the conditional probability of HS as a function of WorldClim hydro-climatological variables in Table 2. The response curves shaded in red represent the top-five environmental determinants for predicting a butterfly’s HS (as permutation importance in Table 2, i.e., when predictions are compared to randomly distributed butterfly occurrences). The environmental determinants are: (**A**) BIO11 = mean temperature of the coldest quarter; (**B**) BIO19 = precipitation of the coldest quarter; (**C**) BIO4 = temperature seasonality (standard deviation ×100); (**D**) BIO5 = maximum temperature of the warmest month; (**E**) BIO10 = mean temperature of the warmest month. In red, the critical tipping points for each climatic feature are highlighted. The blue areas are related to the variability of predicted HS when sampling climatic features.

**Figure 4 entropy-27-00486-f004:**
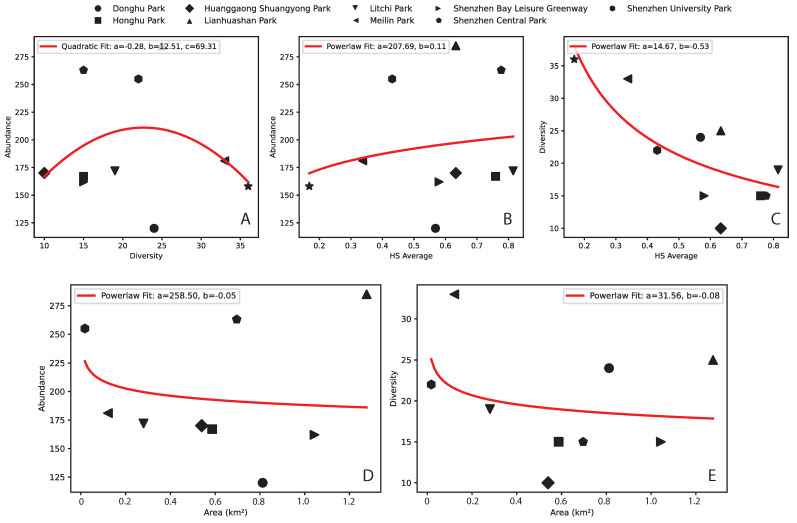
**Butterfly abundance and local richness patterns as a function of habitat suitability and area.** (**A**–**C**) show the abundance–richness, abundance–HS, and species richness–area patterns. (**D**,**E**) show the abundance–area and species richness–area patterns. The abundance is approximated by the total number of observations in parks.

**Figure 5 entropy-27-00486-f005:**
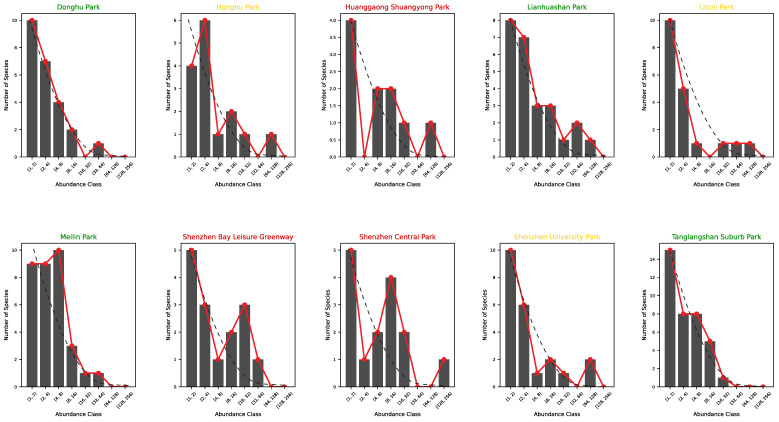
**Preston relative species abundance plots for butterflies in Shenzhen parks.** In black, the raw-data histograms are shown, and in red, the interpolating function connects the maximum values for each abundance class. Honghu, Huanggaong, Shenzhen Central, and Shenzhen Bay Parks show an anomalous bimodal species–abundance pattern reflecting the non-stationarity of butterfly populations (potentially indicative of the ecological communities where butterflies live). The dashed curve refers to the relative optimal Preston plot (log-normal distribution), assumed to fit the Tanglangshan park and used to measure the divergence from other discrete Preston distributions of other parks. Parks are ranked based on the ecosystem fitness index (the EFI in Equation (Equation 9)): the smaller the divergence or RSA and the higher the HS, the higher the EFI (text color from green to red, where green denotes optimal conditions). The EHI is normalized in a [0,1] range as a constructed multivariate index.

**Figure 6 entropy-27-00486-f006:**
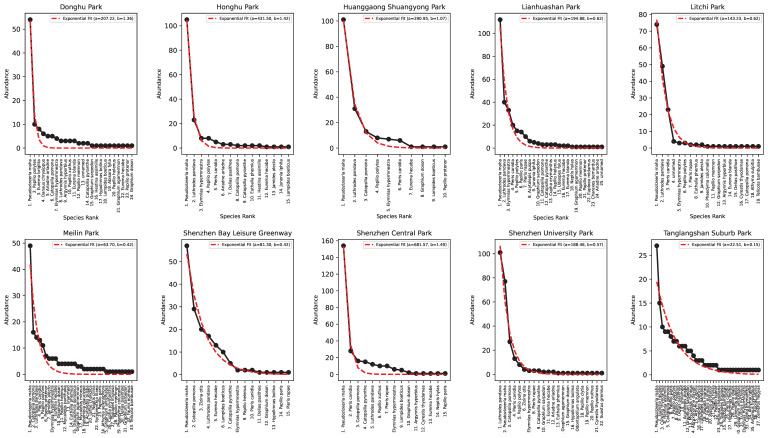
**Abundance rank patterns for Shenzhen parks.** *Pseudozizeeria maha* is the most abundant species for all parks (except for Shenzhen University Park). *Papilio polytes* and *Luthrodes pandava* are very abundant species (second or third rank) across all parks. The species richness of butterflies is quite distinct for each park.

**Table 1 entropy-27-00486-t001:** **Abundance and diversity of butterfly species in monitored parks in Shenzhen**. Here, abundance is considered as the number of butterfly observations across all species. Park details also concern the area, average habitat suitability 〈HS〉, and center-mass coordinates.

Park Name	Abundance	Species Richness	Park Area (km^2^)	〈HS〉	GPS Coordinates (N,E)
Donghu Park	120	24	0.81	0.57	22.588, 114.147
Honghu Park	167	15	0.59	0.76	22.569, 114.12
Huanggaong Shuangyong Park	170	10	0.54	0.63	22.552, 114.059
Lianhuashan Park	285	25	1.28	0.63	22.577, 114.058
Litchi Park	172	19	0.28	0.81	22.546, 114.102
Meilin Park	181	33	0.12	0.34	22.573, 114.036
Shenzhen Bay Leisure Greenway	162	15	1.04	0.58	22.522, 114.021
Shenzhen Central Park	263	15	0.70	0.78	22.551, 114.074
Shenzhen University Park	255	22	0.02	0.43	22.537, 113.931
Tanglangshan Suburb Park	158	36	41.09	0.17	22.574, 114.01

**Table 2 entropy-27-00486-t002:** **Hydroclimatological determinants of butterfly habitat suitability**. The features of precipitation and temperature for the HS in Shenzhen, China, are listed considering their first-order importance and interaction importance, named as percentage contribution and permutation importance (considering all other variables as fixed and changing, i.e., the Sobol total effect of the latter).

Hydroclimatic Variables	Percentage Contribution	Permutation Importance
BIO19: Precipitation of Coldest Quarter	79.1	32.5
BIO4: Temperature Seasonality (standard deviation ×100)	6.5	18.6
BIO11: Mean Temperature of Coldest Quarter	5.2	37.2
BIO6: Min Temperature of Coldest Month	3.8	0.1
BIO13: Precipitation of Wettest Month	1.5	0
BIO3: Isothermality (BIO2/BIO7) (×100)	1	0.6
BIO17: Precipitation of Driest Quarter	0.8	0
BIO9: Mean Temperature of Driest Quarter	0.7	0
BIO1: Annual Mean Temperature	0.4	0
BIO8: Mean Temperature of Wettest Quarter	0.4	0
BIO15: Precipitation Seasonality (Coefficient of Variation)	0.3	0.5
BIO5: Max Temperature of Warmest Month	0.1	8.5
BIO14: Precipitation of Driest Month	0.1	0
BIO7: Temperature Annual Range (BIO5-BIO6)	0.1	0
BIO10: Mean Temperature of Warmest Quarter	0	1.9
BIO18: Precipitation of Warmest Quarter	0	0
BIO16: Precipitation of Wettest Quarter	0	0
BIO12: Annual Precipitation	0	0
BIO2: Mean Diurnal Range (Mean of monthly (max temp - min temp))	0	0

## Data Availability

The raw data are publicly available and their source is referenced in the ”Data” section of the manuscript. Model predictions are available from the corresponding author upon reasonable request.

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
