# Peer review of "Baselining Urban Ecosystems from Sentinel Species: Fitness, Flows, and Sinks"

_entropy, 2025, doi:10.3390/e27050486_

Round 1

Reviewer 1 Report

Comments and Suggestions for Authors

General appreciation:

Although the topic of the article is of interest to both ecologist and land use planers, I was very disappointed with the content. The authors clearly have a good command of mathematical modelling as attested by their previous publications related to human health. Unfortunately, it appears that they do not fully understand key concepts in ecology and conservation biology, and have only very limited and superficial knowledge of the relevant literature. More to the point, their empirical data set (as shown in Table 1) is not of enough quality to allow the authors to provide meaningful results and conclusions. The data set includes only 10 parks, of which 9 have a total area ranging between 0.02 and 1.28 (ha? km²?, this is not indicated in the MS…), whereas a single park has a total surface area of 41.09. This makes a very unbalanced data set for what concerns the influence of area size. Another major point of concern is that the authors do not provide any information on the methods used to estimate species richness. For instance, was the sampling effort the same in all areas? was sampling effort proportional to surface area? etc. I am afraid that although the mathematical modelling might be flawless (although this may require further examination, as results shown in figure 4 suggest some problems with model fitting procedures), the results have absolutely no ecological relevance and add nothing to our understanding of the dynamics of butterfly species diversity in urban environments.

Specific comments   

The English language need some polishing thorough the manuscript. Please avoid unnecessary wordy sentences to maintain conciseness in the text.

The introduction is poorly written. In particular, the first paragraph consists in a series of open questions in what seems to be an attempt to justify the development of a conceptual model.  Instead of that, the authors should have provided a short review of the relevant literature connecting urban ecology, biodiversity and climate change, then point to gaps in knowledge and thus justify their approach.  

Line 49: The authors should explain in some details to what the concept of "ecosystemic fitness of a city" refers.

Lines 68-70: The authors should provide some arguments based on the relevant literature to support their claim that butterflies are "optimal eco-indicators of ecological community fitness and optimal temperature range"

Line 72: Diversity is not equivalent to species richness. This confusion is the major weakness of the article. Species richness, calculated as the total number of species, provides only limited information on diversity. Measures of diversity combine both the number of species and their relative abundances in a composite index.

Lines 79-81:  "Butterflies are potentially excellent eco-indicators for terrestrial ecosystems although they are largely under-reported, and not well considered through their dependencies with ecotones in between waterbodies and hillslopes or urban areas." This is clearly an overstatement, as there exist a relatively important number of scientific articles addressing this question (i.e. Brown & Freitas 2002, J. Insect Conserv.; Kazemi et al. 2022, Landsc. Urban Plan.; Braby et al. 2021, Austral Entomol.; Nelson & Wydowski 2022, J. Lepidopter. Soc.; to cite a few). The authors are encouraged to invest enough time to explore the relevant literature before fully rewriting the introduction of their article.

Lines 88-93: The authors should pay more attention to the content of the article by Kuczynski et al. (2023) and modify the paragraph accordingly. In particular, the increase in diversity with environmental change is mainly relevant for long-lived species. More to the point Kuczynski et al. (2023) point to the problem that apparent increase in species richness with environmental change is largely due to the insufficient length of time series.

Lines 94-95: "Pre-diction (literally meaning ”to mention beforehand” in Latin)": this is not necessary, please remove.

Lines 97-98: please provide references from the relevant literature.

Line 102: " by people to guide conservation efforts [? ? ]": this is just an example of how badly prepared the manuscript was.

Lines 103-109: This seems like verbal intoxication to me. Could the authors substantiate their claims with a decent analysis of the relevant literature?

Lines 246-247: " diversity accounting for abundance which is a disputed indicator defining maximum diversity for evenly distributed species" : This is clearly an unsubstantiated and abusive statement!  

Comments on the Quality of English Language

The English language needs some polishing. There are a few akward sentences and the writing style is unnecessarily wordy in places.  

Reviewer 2 Report

Comments and Suggestions for Authors

Abstract

The abstract uses complex terms like "functional networks" and "attraction basins" without clear definitions, which could confuse readers.

The potential real-world applications of the proposed model for urban planning or climate resilience should be more explicitly stated.

The abstract could briefly mention the analytical methods or data sources used to derive the findings for better clarity on the research approach.

Introduction

The introduction uses complex terms like "eco-flow topology" and "minimum resistance networks" without sufficient explanation; clearer definitions are needed for broader understanding.

The practical implications of the model for urban park design and climate resilience could be more explicitly linked to the research.

Materials and Methods

The explanation of MaxEnt predicting habitat distributions instead of species distributions is intriguing but could benefit from a brief justification on why this approach is preferred for butterfly communities in urban parks.

Including a diagram or flowchart to visually represent ecological flow pathways and the concept of eco-basins would help readers better grasp these complex models and their application.

While the use of AICc for model selection is well-explained, a short description of why this criterion is especially suitable for this study, considering the potential trade-offs between complexity and accuracy, would provide additional clarity.

Results and Discussion

I suggest presenting the results and discussion sections separately, as this will help the reader understand the content more easily.

The concept of butterflies as ecological indicators is fascinating. However, it would be helpful to clarify how the specific environmental pressures mentioned, such as climate or park variability, influence butterfly populations. What specific thresholds or trends in these variables are considered critical for interpreting the butterfly response?"

The analysis of preferential ecological flows and habitat suitability (HS) is insightful. I would be interested in understanding more about how these flows are quantified—are they primarily spatial, or do they also incorporate temporal dynamics?

The relationship between urbanization, park area, and butterfly diversity is intriguing, especially the power-law patterns of abundance and diversity. It might be useful to explore how urbanization factors (such as green space design, pollution, or park isolation) interact with species diversity and HS in more urbanized regions."

Conclusion

It would be better to write a single, coherent paragraph for the conclusion.

While the term bioterraforming is introduced as a key concept for future ecosystem management, its definition and practical implications could be made clearer. The term may need additional explanation or examples to make it accessible to a broader audience.

The conclusion about species richness not being a reliable indicator of ecosystem health is insightful, but it would be beneficial to briefly expand on alternative metrics or indicators that could more accurately reflect ecosystem health in urban environments.

The emphasis on ecotones as critical connectors in urban planning is valuable. Including more specific recommendations or case studies on how cities can integrate this concept into urban design could strengthen the practical applications of the study's findings.

Reviewer 3 Report

Comments and Suggestions for Authors

This is a paper with interesting content but unclear and often convoluted writing. The key aims and methods need to be stated more clearly for readers to understand the content from the beginning. Contributions are unclear in relation to previous work in the field.

Reference formatting does not follow the journal standards.

abstract

The abstract is written in evocative but also confusing language. The initial questions do not do much to give the reader a clear and concise idea of what is to follow.

Initial normative statements need more introduction and grounding, and the abstract needs to establish all jargon used clearly on first mention, such as "EHI". Overall I recommend to cut the abstract in half to concentrate on essential information.

The abstract does not give a clear research question, neither does it give a clear contribution statement.

paper

The content of the paper is great, the topic of the Ecosystem Fitness Indicators timely and relevant, but language is convoluted, sentence structure not clear and often incoherent - a sentence starts out with one thought and ends in another. Even for interested readers this makes reading very difficult. I strongly recommend a review by a scientific writing editor.

The authors should clarify the purpose of the paper much earlier and make this a strong point of the abstract - is the following the key purpose?:

"The model is the focus of this proof-of-concept study that is not an ecological investigation on butterflies nor something applicable only to one species: the paper aims to provide a model for assessing multicriteria fitness, ecological flows, and sinks based on previously developed models."

There is a strong amount of self citing throughout the paper, emphasizing author names, which does not do much to strengthen the argument and credibility of the authors. On the contrary, it gives the impression the authors are seeking to promote their own theories rather than pursuing a study to contribute to the field more broadly. To this end, it would be good to embed the argument the authors are making into some form of structured scientific discourse rather than emphasizing their own previous works. Why is this needed? Which open questions of the scientific community does it answer? What can others learn from this? On page 3 this is slowly developing but comes so late that readers may already have lost interest.

Why are there references to reef occurrences? And images seem to be referred to as pdfs? Please check the scales of the diagrams in Figure 1B.

Instead of consistently mentioning optimizing and optimal, can this be expressed more precisely by naming the specific parameters minimized or maximized? Optimizing implies a value judgment which should not be part of science at least in the analytical stage.

While the theory presented is intriguing, the paper is based on limited data to demonstrate validity of the proposed interpretations of the data. Extrapolating from point data of parks to area based data needed for larger scale ecological networks is not very convincing. This would be OK if the authors did more to critically reflect on their work and state scope and limitations more clearly in the beginning of the paper. Limitations are stated at the end but they seem more to state missing information to complement the author's study rather than a critical reflective stance.

The authors do not account for land use, which is an important factor overlaying the eco-climatic considerations proposed.

For parks, there are further factors that are not considered, such as management and human use / disturbance intensity, etc.

At its core, the paper is an inspired speculation that seemingly deliberately glosses over many factors that could be useful in a more differentiated view of the role of ecohydrological aspects of landscapes in determining fitness landscapes in heterogeneous, urban-dominated environments.

With more embedding in discourse, more critical reflection and more clear writing style and structure the paper could be submitted to a stronger scientific journal.

Comments on the Quality of English Language

needs editing for clarity throughout

Author Response

This is a paper with interesting content but unclear and often convoluted writing. The key aims and methods need to be stated more clearly for readers to understand the content from the beginning. Contributions are unclear in relation to previous work in the field.

Reference formatting does not follow the journal standards.

This is a generic draft format and the whole formatting will be done by the Editorial service after acceptance. This is our experience with the Journal and no issue has been raised by the Journal for this paper. In any event, we polished the manuscript by improving the writing and by making clearer the contribution of our paper with respect what other people have been doing and our own work.

abstract

The abstract is written in evocative but also confusing language. The initial questions do not do much to give the reader a clear and concise idea of what is to follow.

The abstract has been revised by focusing on the model as the main innovation of the work as well as EFI.  

Initial normative statements need more introduction and grounding, and the abstract needs to establish all jargon used clearly on first mention, such as "EHI". Overall, I recommend to cut the abstract in half to concentrate on essential information.

The abstract does not give a clear research question, neither does it give a clear contribution statement.

Thanks. We remade the abstract. EHI was a typo. It is EFI.

The content of the paper is great, the topic of the Ecosystem Fitness Indicators timely and relevant, but language is convoluted, sentence structure not clear and often incoherent - a sentence starts out with one thought and ends in another. Even for interested readers this makes reading very difficult. I strongly recommend a review by a scientific writing editor.

We polished the manuscript by improving the writing and by making clearer the contribution of our paper with respect what other people have been doing and our own work.

The authors should clarify the purpose of the paper much earlier and make this a strong point of the abstract - is the following the key purpose?:

"The model is the focus of this proof-of-concept study that is not an ecological investigation on butterflies nor something applicable only to one species: the paper aims to provide a model for assessing multicriteria fitness, ecological flows, and sinks based on previously developed models."

Yes, the model was developed to assess the multicriteria ecoclimatological fitness, ecological flows, and sinks of ecosystems. An application of the model was done for butterflies in urban settings.

The emphasis on the eco-hydro-climate features exerted by climate factors, park-presence and connectivity. This emphasis is due to the interest of the group into structural ecohydrology as the main determinant of ecosystem function. These features are used to formulate EFI whose distribution is non-random and following features of HS that are typically not provided by Species Distribution Models. These inferred features are ecological corridors and eco-basin boundaries that are really relevant for ecosystem diagnosis and planning, because informing about likely species connectivity (flows) in response to environmental pressure and the confinement of this connectivity.

The model is an open box for handling any eco-environmental data at any higher resolution. It is certainly not a closed black box that works only with the ingredients we used in this paper. Therefore, the model can be applied to be more practical for sure.

There is a strong amount of self citing throughout the paper, emphasizing author names, which does not do much to strengthen the argument and credibility of the authors. On the contrary, it gives the impression the authors are seeking to promote their own theories rather than pursuing a study to contribute to the field more broadly. To this end, it would be good to embed the argument the authors are making into some form of structured scientific discourse rather than emphasizing their own previous works. Why is this needed? Which open questions of the scientific community does it answer? What can others learn from this? On page 3 this is slowly developing but comes so late that readers may already have lost interest.

Very frankly, our aim was precisely to cite parts of the model developed in the past in previous papers and frame that into the current scientific aim and application. It was not a form of gratuitous self-referencing but a way to build up a cohesive and connected theory for how our model building blocks work together. In particular, this is for the approach of the Optimal Information Network model adopted on the MaxEnt model predictions (accounting for non-linearity), to extract the ecological corridors and flows. This is equivalent

In any event, we reduced the number of self-references and critically pinpointed our attention to the most salient model’s parts.

Why are there references to reef occurrences? And images seem to be referred to as pdfs? Please check the scales of the diagrams in Figure 1B.

Apologies for these typos. It should be butterflies of course.

Instead of consistently mentioning optimizing and optimal, can this be expressed more precisely by naming the specific parameters minimized or maximized? Optimizing implies a value judgment which should not be part of science at least in the analytical stage.

We disagree that the value-judgment is not a part of the scientific discourse. Firstly, evolution make unconscious and automated value ‘’judgements’’ over competing criteria (see https://www.weizmann.ac.il/mcb/alon/download/pareto-task-inference-parti-method). Second, optimality here is in the context computational complexity that deals with extracting the most informative set of data with the minimum effort, i.e. model design. This is a standard practice in modeling and biology alike. This is what the following sentences mean.

‘’optimal trade-off between model complexity (defined as the number of environmental variables used as predictors) and model accuracy (that is the distance between predictions and data) ‘’

‘’ The optimal models were selected using Akaike's Information Criterion, which was corrected for small sample sizes ($\Delta$AICc = 0), this approach penalizes excessive model complexity and facilitates the selection of models with an optimal number of parameters\citep{tobon2022incorporating}.’’

To explain further we emphasized the following related to our Optimal Information Network model:

‘’ Optimal Ecological Networks} (OEN) -- that are the most predictive networks of observed ecological patterns (see \citet{li2021inferring}) considering data uncertainty and systemic information -- can be defined as thresholded maximum gradient pathways as’’

As for optimality about ecological patterns we already mentioned the following ‘’ (i.e., the lognormal Preston plot of the Relative Species Abundance dependent on the optimal organization of habitats, that is also the most predictive because non-randomly organized).’’ and we made it clearer that is dependent on the optimal ecological network organization of species as a function of green habitats.

While the theory presented is intriguing, the paper is based on limited data to demonstrate validity of the proposed interpretations of the data. Extrapolating from point data of parks to area based data needed for larger scale ecological networks is not very convincing. This would be OK if the authors did more to critically reflect on their work and state scope and limitations more clearly in the beginning of the paper. Limitations are stated at the end but they seem more to state missing information to complement the author's study rather than a critical reflective stance.

We changed the intro by including the meaning of inference in this and other papers in the context of ecosystem engineering and design. We also suggest the reviewer to consider our answers to other reviewers’ comments.

‘’ Species as Indicators of Ecosystem Fitness Stressed by Climate Shifts: Inferring Magnitude and Pathways’’

‘’ From outputs (e.g. eco-patterns and their change, i.e. systemic stress) the aim is to recon- struct the input determinants probabilistically: our work is focused on eco-networks and flows (that is considering how the inputs are structured), and their change from optimal, random or alternative known networks. Inverse modeling knows the causal function between inputs and outputs, and that function is used in the inversion to calculate the unknown parameters linking input-output changes. Inverse modeling is often used when the forward model (from cause to effect) is known, and we want to reverse it; such as through Monte Carlo filtering in global sensitivity and uncertainty analyses. Inference, instead, is a data-based reconstruction of structure-function causal relationships (of causes and effects, such as climate and species) that cannot be know a priori or modeled by mechanistic equations, therefore useful to derive analytics of the critical and predictable dynamics, indicators, and system response intensity. Thus, the computing (e.g. via MaxEnt) that is a low-level machine-learning model) is not just about learning the parameters but learning all functional forms. Learning the precise physics is particularly daunting and maybe utopian for highly complex systems with many components/variables interacting non-linearly (due to space-time delays and as stress-strain non-linear response). The inferred networks, because of the inference construction maximiz- ing traceability, are the most predictive for the observed ecological patterns. However, we also aim to give causal physicality to the inferred networks: e.g. networks encode ecohydro- logical networks such as species dispersal networks and hydrological flow networks (in natural landscapes and green corridors) that are dependent on each other and impacting climate.

In this modeling framework, minimum resistance networks (or maximum information flow/maximum prediction as in the Optimal Information Network model of Servadio and Convertino (2018)
and Li and Convertino (2021)) along the maximum gradient of a multicriteria function (defining the Habitat Suitability, HS) allow to identify paths where species can spread (as ”waves”) in a suitable climate range. These networks have the highest predictability of ecological pat- terns, such as HS. ‘’

The authors do not account for land use, which is an important factor overlaying the eco-climatic considerations proposed. For parks, there are further factors that are not considered, such as management and human use / disturbance intensity, etc.

All these elements are certainly potentially relevant; however, this paper is proposing a model with a minimal information for two reasons: (1) the core is the model itself (inferring ecological networks from HS as the Digital Ecological Model in analogy to extracting hydrological networks from Digital Elevation Models); and (2) the second aims is inferring HS and HS networks (in the simplest way) where the focus is only on hydro-climatic features and their potential robust predictability (potential causality embedded into the ecohydrological physics) on ecological communities in parks. The overarching goal is to improve later on this model with more detailed analysis on which species is better suited to be representative of ecological communities and what are the park features and surrounding features that matters. The list can be huge and should be bounded to the scope of the analysis, like we did here: the model.  

At its core, the paper is an inspired speculation that seemingly deliberately glosses over many factors that could be useful in a more differentiated view of the role of ecohydrological aspects of landscapes in determining fitness landscapes in heterogeneous, urban-dominated environments.

The most important aspect that is overlooked is that what we targets are predictions, and more precisely backward predictions or inference. All what we talk about are inferred relationships without any assumptions into the modeling engine at the basis of our work.

Our ‘’speculations’’ (I would say discussion from the results of the inference) are discussions about something physical (eco-hydroclimatological response patterns and suitability) that is known and recognized to exist and very important for the emergence of ecological function (see Rodriguez-Iturbe et al. 2009 ‘’ River networks as ecological corridors: A complex systems perspective for integrating hydrologic, geomorphologic, and ecologic dynamics’’); however, this systemic function is very hard to measure directly at the species scale. That is why, in our perspective as engineers (not as basic scientists), we developed an inferential model to support the quantification of the potential community response to hydroclimatic pressure and its gradients as eco-pathways of change.

The potential species movement, that is hard to measure, is a reflection of the ecohydrological structure of landscapes, and we are just modeling it (or better propose a model to model it). Further effort will investigate how much this reflect ‘’true’’ response and flows; however, macroscopic assessment like ours are grounded on the physics of the phenomena we are experts on (ecohydrology) and this makes the machine learning models we used explainable models. The information we are dealing with is physical because it deals with flows of matter (species), senses, and energy in response to environmental triggers (see Brose et al., 2025; Little et al., 2022; Suzuki et al., 2022). Additionally, this information is really needed because it is important to assess urban ecosystem response to the hydroclimatic pressure mediated by the environmental configuration (see our recent paper Wu et al., 2025 ‘’ Ecological Corridor Design for Ecoclimatic Regulation: The South China Tiger’’). We do so by inferring how much climate is encoded into butterflies modulated by parks. Information is physical into the ecohydrological structure of ecosystems, yet physical in terms of decision-making impacts.

With more embedding in discourse, more critical reflection and more clear writing style and structure the paper could be submitted to a stronger scientific journal.

Thanks. We revised the manuscript. Considering the amount of work we needed to do for now we stick with Entropy.

Reviewer 4 Report

Comments and Suggestions for Authors

Dear Authors,

The research is well designed, and the manuscript is well-written. I enjoyed learning and reading your work very much.

There are only some minor comments which I hope you will address and improve its original impact of this work:

1) The current section 4.1 includes the innovations for ecological applications/implications and methodology. I think it would be great to separate them explicitly. For example, the current three main features can be the innovations for ecological applications/implications. I will talk about the methodological innovations below.

2) Since modeling habitat suitability with many innovations has been conducted widely in the modeling community, I would like to ask you to discuss how your work has difference from or advantage on current published work in section 4.1. I hope the discussion can be richer. Some relevant work: (1) https://doi.org/10.1016/j.ecolmodel.2017.04.005, (2) https://doi.org/10.1016/j.ecolmodel.2024.110909.

When you revise the manuscript, please scope up your current work, for example, how do your methods and application advance the current study? Or, what other discipline contexts beyond your model/example might your general advice or lessons apply to?

3) The resolution of Figures 3, 5, and 6 should be improved. It is difficult to read.

4) I found it was difficult to link the text and Figures 5 and 6. Authors should improve the connection/description for these two figures. Moreover, Figures 5 and 6 include information of parks. The location information should be displayed on a detailed map. Perhaps improve Figure 2A format to indicate the locations of parks.

5) Please introduce all indexes criteria. For example, what is the range of Ecosystem Health Index value considered as healthy? Authors should mention this first before evaluating each park.

Author Response

Dear Authors,

The research is well designed, and the manuscript is well-written. I enjoyed learning and reading your work very much.

There are only some minor comments which I hope you will address and improve its original impact of this work:

1) The current section 4.1 includes the innovations for ecological applications/implications and methodology. I think it would be great to separate them explicitly. For example, the current three main features can be the innovations for ecological applications/implications. I will talk about the methodological innovations below.

We thank the reviewer for the comment; indeed, we believe that the separation would is beneficial to highlight how the methodology innovation is much more generalizable and yet can have positively impactful applications for any species and ecosystems. The specific butterfly and SZ data have been separated for stakeholders. Therefore, we further polished the section by separating the two elements.

2) Since modeling habitat suitability with many innovations has been conducted widely in the modeling community, I would like to ask you to discuss how your work has difference from or advantage on current published work in section 4.1. I hope the discussion can be richer. Some relevant work: (1) https://doi.org/10.1016/j.ecolmodel.2017.04.005 , (2) https://doi.org/10.1016/j.ecolmodel.2024.110909.

This has been widely stated we believe. The main innovation is the inference of ecological networks, basins, and sinks. These ecological networks are both predictive networks and physical networks representing the climate-based minimum resistance to species flows. We cited the papers you suggested.

When you revise the manuscript, please scope up your current work, for example, how do your methods and application advance the current study? Or, what other discipline contexts beyond your model/example might your general advice or lessons apply to?

MaxEnt has been developed and applied by my group in predicting hydroclimatological hazards such as landslides and floods, however without inferring functional networks. Further uses were also mentioned in the following paragraph.

‘’ predicted habitat hydroclimatic flood and landslide risks in previous studies, respectively; however, without inferring the underlying functional networks but looking into the contribu- tion of hydrologic networks to the risks. Ecological networks among species were previously inferred, by using maximum entropy approaches, on Relative Species Abundance data, such as by Li and Convertino (2021a) for fish and by Galbraith et al. (2022) for the ocean mi- crobiome, and used with Graph Neural Networks for forecasting bioaquatic risks (Wang and Convertino, 2023) such as algal blooms. ’’

3) The resolution of Figures 3, 5, and 6 should be improved. It is difficult to read.

The resolution has been improved, although we are not sure why the visualization looked with low resolution for the reviewer. This is the best we can do considering the export function in ArcGIS.

4) I found it was difficult to link the text and Figures 5 and 6. Authors should improve the connection/description for these two figures. Moreover, Figures 5 and 6 include information of parks. The location information should be displayed on a detailed map. Perhaps improve Figure 2A format to indicate the locations of parks.

The name of the parks and their location is contained in Figure 2. We increased the size of the fonts to the extent possible. However, Fig. 6 will be an online figure (as all others due to the fact that Entropy is online only), yet people can zoom in. The species are too many and the names are too long. It is hardly possible to increase the size of fonts. The key of figures is the patterns.

5) Please introduce all indexes criteria. For example, what is the range of Ecosystem Health Index value considered as healthy? Authors should mention this first before evaluating each park.

There is no upper bound into EFI since it is a constructed index like risk or other indicators like FLII and EHI proposed in our paper for Blue Carbon Ecosystems (Zhang and Convertino, 2014). However, in our context we proposed a normalized version of the index considering the maximum value of EFI and rescaling all other values. Of course, EFI can be normalized in different ways, but the normalization does not imply anything physical; however, when comparing ecosystems the same normalization should be adopted.

We revised the following paragraphs:

‘’EHI is normalized in a [0,1] range as a constructed multivariate index.’’

‘’In this perspective, ecosystem health (or fitness) should not be interpreted only as related to the ecological dysbiosis of species and habitats (e.g., of symptoms), but about the ecological function of ecosystems. The latter is associated with the robust ecohydrological determinants of ecosystem function, leading to optimal or de- sired patterns that likely increase ecosystem services (see Convertino and Valverde Jr (2019)). This is for instance the case of species speciation and dispersal leading to biodiversity pat- terns and cascading ecosystem services. The core ecohydrological structure of ecosystems, as corridors and flows, is the basis for the healthy/functional feedback of all species, communities, and climate (Wang and Convertino (2023) and (Zhang and Convertino, 2024)).

‘’Ecosystem health is not just predicting diseases in populations and communities, but also mapping and engineering the baselining eco-environmental foundations of all species and habitat functions. This effort deliberately aims to increase the ecohydrological fitness of ecosystems in a way that positive species-habitat-climate feedback are reinforced, and negative cascading outcomes such as biodiversity loss and hydrological extremes are minimized (Funabashi, 2018; Convertino

and Valverde Jr, 2019; Funabashi, 2024).’’

Round 2

Reviewer 1 Report

Comments and Suggestions for Authors

I have read the revised version of the MS and the detailed answers provided by the authors to the refrees' comments. Overall, the MS has been largely improved. However, as an ecologist, I think that the limitations I pointed to in my first evalution do remain to a large extent. I do understand that the authors are more interested in proposing a new method than in providing results of biological/ecological significance on butterfly fish. In that respect I wonder why the authors chose to use a real (but very limited) data set instead of relying on a virtual one. In essence, I am not qualified to assess the merit of the article from a modelling point of view. I can only reaffirm that in my opinion the results are of very little importance or relavance for management or conservation given the limitations associated with the original data set. 

Author Response

I have read the revised version of the MS and the detailed answers provided by the authors to the referees' comments. Overall, the MS has been largely improved. However, as an ecologist, I think that the limitations I pointed to in my first evaluation do remain to a large extent. I do understand that the authors are more interested in proposing a new method than in providing results of biological/ecological significance on butterfly fish. In that respect I wonder why the authors chose to use a real (but very limited) data set instead of relying on a virtual one. In essence, I am not qualified to assess the merit of the article from a modelling point of view. I can only reaffirm that in my opinion the results are of very little importance or relevance for management or conservation given the limitations associated with the original data set. 

People are driven by their mental model. We respect the position of the reviewer that is reflecting the world and mental models of ecologists. We are engineers and we submitted the journal to Entropy, that is not quite an ecology journal but a journal focused on information theory, statistical physics and their applications, and ranked in that category as a Q1 journal. Therefore, our model is presented as an innovative model based on entropy because it is based on information theory, and applied to butterflies as a case study due to previously stated reasons. We do not claim to investigate the basic ecology of butterflies nor to incorporate all elements of butterfly’s ecology that are relevant for butterfly conservation. We also agree that further space-time presence data (and abundance perhaps) is necessary to have high-resolution information for targeted habitat management and conservation. However, here we present the model to handle that sort of information, the model for handling the whole complexity of the problem to which the reviewer is interested. The model is an open box for handling any eco-environmental data at any higher resolution. It is certainly not a closed black box that works only with the ingredients we used in this paper. Therefore, the model can be applied to be more practical for sure.

The second aspect is the emphasis on the eco-hydro-climate features exerted by climate factors, park-presence and connectivity. This emphasis is due to the interest of the group into structural ecohydrology as the main determinant of ecosystem function. These features are used to formulate EFI whose distribution is non-random and following features of HS that are typically not provided by Species Distribution Models. These inferred features are ecological corridors and eco-basin boundaries that are really relevant for ecosystem diagnosis and planning, because informing about likely species connectivity (flows) in response to environmental pressure and the confinement of this connectivity. This information is physical or at least predictive in a broader information dynamics theory (see the paper ‘’ Dynamic patterns of information flow in complex networks’’).

We further mention these aspects in the paper and we would be grateful if this perspective, driven by our main mental model and objectives is welcomed. Further discussions are ongoing with local ecologists, environmental scientists, hydrologists, engineers, and decisions makers for the high-resolution use of this model at different scales.

Reviewer 2 Report

Comments and Suggestions for Authors

Accept in present form

Author Response

We really thank and appreciate the reviewer for welcoming our paper after our improvements.

Reviewer 4 Report

Comments and Suggestions for Authors

Dear Authors,

Thank you for addressing my comments accordingly and appropriately.